# Health seeking behavior after the 2013–16 Ebola epidemic: Lassa fever as a metric of persistent changes in Kenema District, Sierra Leone

Mikaela R. Koch[1]*, Lansana Kanneh[2], Paul H. Wise[2], Lianne M. Kurina[1], Foday Alhasan[3], Robert F. Garry[4,5], John S. Schieffelin[6], Jeffrey G. Shaffer[7]*, Donald S. Grant[3,8]*

**1** Program in Human Biology, Stanford University, Stanford, California, United States of America, **2** Pediatrics–Neonatal and Developmental Medicine, Stanford University, Stanford, California, United States of America, **3** Viral Hemorrhagic Fever Program, Kenema Government Hospital, Kenema, Sierra Leone, **4** Tulane University, School of Medicine, Department of Microbiology and Immunology, New Orleans, Louisiana, United States of America, **5** Zalgen Labs, LCC, Germantown, MD, United States of America, **6** Sections of Infectious Disease, Department of Pediatrics, School of Medicine, Tulane University, New Orleans, Louisiana, United States of America, **7** Department of Biostatistics and Bioinformatics, Tulane School of Public Health and Tropical Medicine, New Orleans, Louisiana, United States of America, **8** Ministry of Health and Sanitation, Freetown, Sierra Leone

* mikakoch@alumni.stanford.edu (MRK); jshaffer@tulane.edu (JGS); donkumfel@yahoo.co.uk (DSG)

**Data Availability Statement:** All relevant data are within the manuscript and its Supporting Information files.

## Abstract

### Background

The West African Ebola epidemic of 2013–2016 killed nearly 4,000 Sierra Leoneans and devastated health infrastructure across West Africa. Changes in health seeking behavior (HSB) during the outbreak resulted in dramatic underreporting and substantial declines in hospital presentations to public health facilities, resulting in an estimated tens of thousands of additional maternal, infant, and adult deaths per year. Sierra Leone's Kenema District, a major Ebola hotspot, is also endemic for Lassa fever (LF), another often-fatal hemorrhagic disease. Here we assess the impact of the West African Ebola epidemic on health seeking behaviors with respect to presentations to the Kenema Government Hospital (KGH) Lassa Ward, which serves as the primary health care referral center for suspected Lassa fever cases in the Eastern Province of Sierra Leone.

### Methodology/Principal findings

Presentation frequencies for suspected Lassa fever presenting to KGH or one of its referral centers from 2011–2019 were analyzed to consider the potential impact of the West African Ebola epidemic on presentation patterns. There was a significant decline in suspected LF cases presenting to KGH following the epidemic, and a lower percentage of subjects were admitted to the KGH Lassa Ward following the epidemic. To assess general HSB, a questionnaire was developed and administered to 200 residents from 8 villages in Kenema District. Among 194 completed interviews, 151 (78%) of respondents stated they felt hospitals

**Funding:** Supported by National Institute of Allergy and Infectious Diseases grants/contracts AI104621, AI114855, AI115754, U19AI135995, and U01AI151812 (to LK, FA, JSS., RFG., JGS, and DSG) and by a Stanford University Undergraduate Research Grant (to MRK). The funders had no role in study design, data collection and analysis, decision to publish, or preparation of the manuscript.

**Competing interests:** I have read the journal's policy and the authors of this manuscript have the following competing interests: L.K., F.A., J.S.S., R. F.G., J.G.S., and D.S.G are members of the Viral Hemorrhagic Fever Consortium (www.vhfc.org). The VHFC is a partnership of academic and industry scientists who are developing diagnostic tests, therapeutic agents, and vaccines for Lassa fever, Ebola, and other severe diseases. Tulane University and its various academic and industry partners have filed U.S. and foreign patent applications on behalf of the consortium for several of these technologies. Technical information may also be kept as trade secrets. If commercial products are developed, consortium members may receive royalties or profits. This does not alter our adherence to all policies of the NIH and PLOS Neglected Tropical Diseases on sharing data and materials. Financial and non-financial competing interests that the editors consider relevant to the content of the manuscript have been disclosed. R. F.G. is a co-founder of Zalgen Labs, LLC. All other authors declare no competing interests.

were safer post-epidemic with no significant differences noted among subjects according to religious background, age, gender, or education. However, 37 (19%) subjects reported decreased attendance at hospitals since the epidemic, which suggests that trust in the healthcare system has not fully rebounded. Cost was identified as a major deterrent to seeking healthcare.

## Conclusions/Significance

Analysis of patient demographic data suggests that fewer individuals sought care for Lassa fever and other febrile illnesses in Kenema District after the West African Ebola epidemic. Re-establishing trust in health care services will require efforts beyond rebuilding infrastructure and require concerted efforts to rebuild the trust of local residents who may be wary of seeking healthcare post epidemic.

### Author summary

The West African Ebola epidemic of 2013–2016 killed nearly 4,000 Sierra Leoneans and devastated health infrastructure. There is limited information regarding the effects of the epidemic on health seeking behavior. Lassa fever is clinically similar to Ebola and is prevalent in Kenema District, located in the Eastern Province of Sierra Leone. There was a significant decline in the number of individuals seeking care for suspected Lassa fever following the epidemic. Responses to a questionnaire completed by 194 Sierra Leonean residents suggested that confidence in the healthcare system was not fully restored. Cost was identified as a major deterrent to seeking healthcare. Additional community sensitization is needed to convey the need, and importance, of seeking care for Lassa fever.

## Introduction

The largest known Ebola epidemic in history began in 2013 ultimately spanning over 30 months and killing over 11,000 people in Guinea, Sierra Leone, Liberia and other countries [1,2]. The epidemic was reportedly fueled and perpetuated by misinformation. Erroneous messaging, rumors, conspiracy theories—including that Ebola virus (EBOV) was a hoax and that consumption of bush meat was sustaining the outbreak—and predictions by government advisors that the outbreak would soon burn out, contributed to the public health disaster [3–5]. Healthcare facilities, which were plagued by large numbers of heath care worker (HCW) deaths, became known by some as places where people went to die, and thus many individuals stopped seeking care altogether [6–10]. The long-term effects that a traumatic event, such as the West African Ebola epidemic, pose to trust in the healthcare system are not well-documented in the scientific literature.

While Ebola was directly responsible for thousands of deaths in Sierra Leone (approx. 4000), changes in health seeking behavior (HSB) due to documented fear, misinformation, distrust, and a reduction in medical services, resulted in considerable non-Ebola mortality that continues to this day [11–14]. Maternal and child health services were particularly affected [10]. At the height of the epidemic, there was a 31–37% drop in facility-based child deliveries and an 80% decrease in pediatric admissions for malaria in Sierra Leone [15]. Such reduction in healthcare provision was predicted to result in an additional 4,022 maternal deaths per year

and 6,700 additional infant deaths per year in the region [10]. Additionally, the number of major surgeries conducted, in the region, during the West African Ebola epidemic decreased by 50% [16,17]. Beyond the barriers present and inherent to the region (limited health service availability, poverty, lack of educational access), the epidemic may have left a lasting impact on the perceptions of, and attitudes towards, the healthcare system [18–20]. Health seeking behavior, as assessed in this study, is defined as "a sequence of remedial actions individuals undertake to rectify perceived ill health" [21]. The underlying assumption is that health is influenced by behavior and behavior itself is subject to modification, as influenced by external and internal factors [22]. Proxies used for health seeking behavior in the literature include, healthcare consumption patterns, provider visits, and self-reported care seeking [23–26].

In the wake of the Ebola epidemic, there has been significant investment and attention in health systems strengthening for the affected countries of West Africa [27,28]. In April 2015, the Sierra Leone MoHS detailed a strategy to strengthen the health system which included, as its first key areas, establishing safe work settings and increasing the workforce [29]. NGO's in the area, such as Partners in Health, joined this objective working to improve the infrastructure of local health facilities [27]. In Kenema specifically, a new 48 bed Lassa fever ward was built following the epidemic [30]. While the desire to strengthen the health system is often the 'dominant narrative' after an epidemic, whether individuals are availing themselves of these services has not been adequately addressed in this setting [31]. This study sought to comprehend that knowledge gap by 1) assessing HSB of patients with suspected Lassa fever and 2) assessing general HSB of individuals through use of a questionnaire, in the Kenema district of Sierra Leone, a region that was severely impacted by Ebola. Based on the incredible shock to the system (societal, healthcare, governmental, economic etc.) and clear evidence of a dramatic change in health seeking behavior during the epidemic across the nation, the authors hypothesized that there would remain an impact on hospital attendance for Lassa fever observed in the population years after the epidemic concluded.

Lassa fever and Ebola are both hemorrhagic fevers—severe acute viral infections induced by Lassa virus (LASV), an arenavirus, and EBOV, a filovirus, respectively [1,32]. Patients present with similar symptoms, including fever, malaise, vomiting, and bleeding. While many individuals infected with LASV do not present to a health care provider or facility, both prior to, and following the epidemic, those that do are often misdiagnosed as having influenza or malaria given the initial nonspecific symptoms and aggressive local treatment programs for malaria [32–35]. LASV is transmitted through contact with *Mastomys natalensis*, the multi-mammate mouse, and other rodent reservoirs [36,37]. Human to human transmission of LASV can also occur through contact with bodily fluids, similar to EBOV, although less frequently [38–40]. The incubation period of Lassa fever is 6–21 days and symptoms may be mild and subclinical [35,41,42]. Severe cases progress to multi-organ failure and internal hemorrhaging. For those requiring hospitalization, the suspected case fatality rate is variable—between 20% in Nigeria to 70% in Sierra Leone [43,44]. While there are no precise data on the incidence or prevalence of Lassa fever, it has been estimated that there are tens of thousands of deaths each year in West Africa [45]. The case fatality rate among hospitalized Lassa fever (LF) patients has been reported to be 69% [34].

The first goal of this study was to measure potential changes in HSB for suspected Lassa fever. Given the similarity in the diseases, HSB for Lassa fever may serve as a metric for the impact of the West African Ebola epidemic on HSB. The second goal of this study was to evaluate attitudes toward healthcare in a Lassa fever endemic region to identify barriers (e.g cost, time, information, trust etc.) and provide insights into whether there was a persistent effect of the epidemic on communities' relationship to, and perception of, their healthcare system.

## Methods

### Ethics statement

Compilation of surveillance data regarding patients presenting to Kenema Government Hospital (KGH) with suspected Lassa fever was approved by the Sierra Leone Ethics and Scientific Research Committee. Informed consent was obtained from suspected Lassa fever subjects presenting to KGH or one of its public health units or non-identifiable presentation frequencies were captured as part of routine hospital surveillance activities. Administration of health seeking behavior (HSB) questionnaires was approved by the Sierra Leone Ethics and Scientific Research Committee and the Institutional Review Board of Stanford University (#45598). All subjects who participated in the HSB questionnaire and/or their legal guardians provided written informed consent. Consent was obtained for each participant in the presence of a witness. Illiterate participants consented with a thumbprint rather than a signature. Given Mende, the most common local language, is written and read by very few, all consents and questionnaires were administered orally in the resident's preferred local dialect (Krio or Mende). As per the standard outreach team practice, upon entering the village, permission from the village leaders were obtained. All methods were carried out in accordance with relevant guidelines and regulations, including the Declaration of Helsinki.

### Lassa fever surveillance database

Surveillance and demographic data on suspected Lassa fever cases have been captured using a suite of case report forms (CRFs) and log books at KGH since 2006. These data were captured from patient referral CRFs, pre-admission evaluation CRFs, patient medical charts (for admitted subjects), and a paper-bound log book maintained at the KGH Lassa Ward. Data for subjects presenting to KGH (regardless of admission status) were verified against pre-admission evaluation CRFs, and data for admitted subjects were verified against patient medical charts. Patient survival outcome was considered with the hierarchical order of validity: 1) Lassa Ward log book; 2) pre-admission evaluation CRF; 3) patient referral CRF; and 4) patient medical chart and nurses notes. Data were managed and linked across data sources using the SAS statistical package (version 9.4, SAS Institute, Cary, NC). The complete data capture and management process for suspected Lassa fever cases at KGH is described in further detail by Shaffer et al. [46].

### Database analysis

For this study, the number of annual visits by age, gender, and survival outcome were abstracted from the Lassa fever surveillance database from 2011–2019. During the height of the West African Ebola epidemic (between May 24, 2014 and January 26, 2015), resources were devoted to containing the outbreak, and the data capture processes were adapted for incoming Ebola subjects. During this time, data for Lassa cases was sporadic and unreliable due to limited human resources available for Lassa fever surveillance; therefore, a detailed analysis on Lassa presentations during this time-period is not covered in this work.

### HSB questionnaire

The questionnaire used in this study was adapted from a questionnaire on general health seeking behavior employed by Medair and Mercy Corp in Uganda, along with demographic data including age, gender, village of residence, and education [18]. Four specific study pertinent questions were added. The HSB questionnaire was provided in the two major written languages used in the villages, English (S1 Table) or Krio (S2 Table), but given low literacy rates,

was delivered orally in Krio or Mende. Questionnaires were numbered and did not contain any personal identifiers. The villages were selected to represent a cross-section of the district based on distance from KGH (ranging from 22–51 km away) and population size (ranging from 600 to 2100 individuals). The outreach team had contacted each village in previous years either for case identification, educational programming directed at Lassa fever awareness, or for participation in Viral Hemorrhagic Fever Consortium (VHFC) research studies. The twenty questions took on average of ten minutes to complete. Overall, 200 participants were interviewed.

## Statistical analysis

Surveillance data included both categorical and continuous responses and were presented as frequencies and percentages, or means and standard deviations, respectively. Data were classified in terms of pre-Ebola (2011–2013); Ebola (2014–2016); and post-Ebola (2017–2019) time periods according to data of clinical presentation to a hospital or public health unit in Sierra Leone. Nonparametric approaches (Fisher's Exact and Kruskal-Wallis tests) were used to account for departures in diagnostics for their analog parametric approaches (Pearson's chi square and ordinary one-way analysis of variance). The logistic regression approaches were applied due to the categorical nature of the data captured in the HSB survey. The Type 1 error threshold was set at 5%, and those p-values less than this threshold were considered as statistically significant. Statistical significance focused on differences observed between pre- and post-Ebola time periods (2011–13 and 2017–19, respectively) that were not due to chance alone. Significance tests were performed using the SAS statistical application (version 9.4, Cary, NC).

General HSB questionnaire responses (questions 1, 2, & 4–12) were analyzed to identify potential predictors of responses to the specific study pertinent questions (3 & 13–15). The analysis was performed by running logistic regression models in STATA v. 15.1 (Stata Corp, College Station, Texas). The logistic regression analyses were performed using full main effects (no interaction) models, where outcomes were dichotomized and modeled against subject age, gender, religion, education, and village of residence. For this analysis, age categories, 18–27, 28–36, 37–45, and 45+, were selected to accommodate the age inclusion criteria for the HSB questionnaire. All logistic regression analysis were run with Village 1 as the reference village. Village 1 was selected to be the reference village as it was one of two villages with the most respondents. One of these two villages was arbitrarily chosen as the reference village. When appropriate, responses were contracted to dichotomous measures. All chi-square tests and trends were analyzed using SAS/STAT software version 9.4 (Cary, NC). All statistical tests applied a type I error threshold of 5%.

## Results

### Persistent decline in suspected Lassa fever cases following the West African Ebola epidemic

In the three-year period (2017–19) following the West African Ebola epidemic, as compared to the three years prior to the epidemic, there was a decline in the numbers of subjects presenting to the Kenema Government Hospital (KGH) Lassa ward with suspected Lassa fever and the numbers of patients admitted to the KGH Lassa ward (Fig 1, S3 Table). The average annual number of suspected cases of Lassa fever presenting to KGH was 698 from 2011 to 2013, the three years before the beginning of the epidemic (Fig 1A). During the epidemic an average of 473 suspected cases presented, although, as noted, data collection was suboptimal during this period. In the three years following the epidemic, an average of 326 suspected cases presented

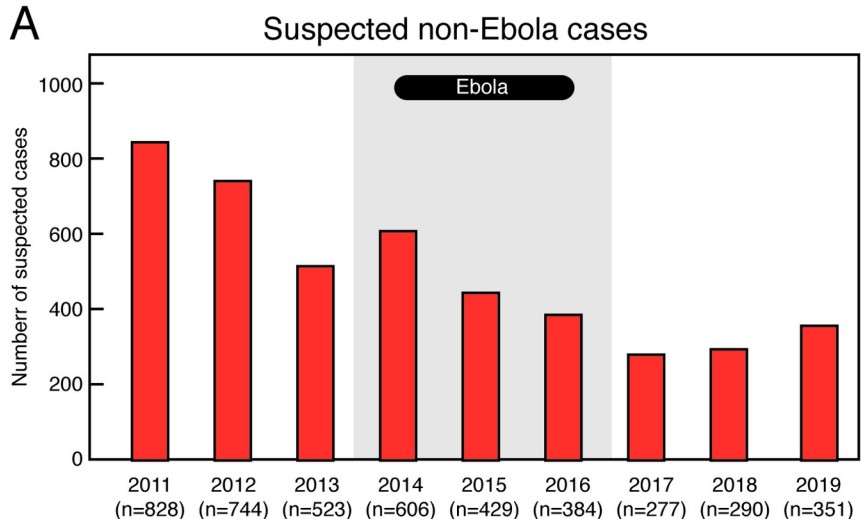

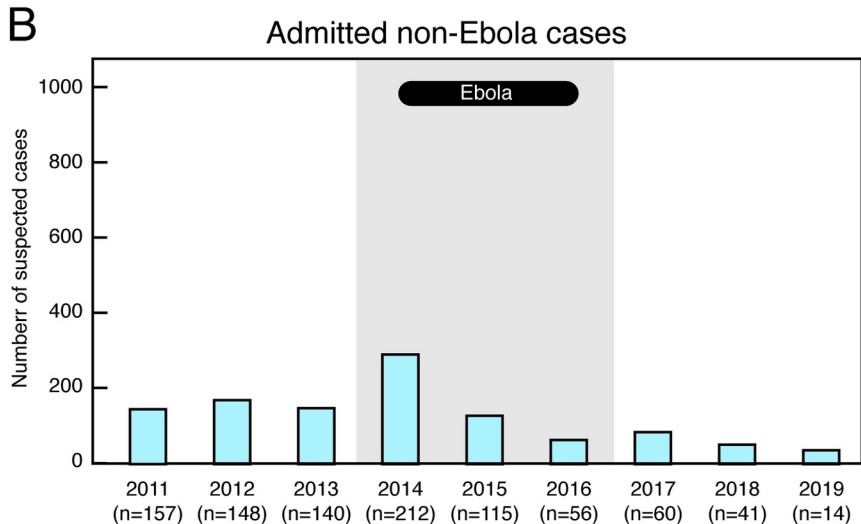

**Fig 1. Annual numbers of subjects presenting and admitted to the Kenema Government hospital Lassa ward with suspected Lassa fever.** Panel A. Subjects presenting with suspected Lassa fever from 2011–19. Panel B; Patients admitted to the Lassa Ward from 2011–19. Grey shading indicates the years 2014–16 during the West African Ebola epidemic.

to KGH (Fig 1A). The proportion of cases post-epidemic was statistically significantly lower than the proportion pre-epidemic (p = .049, Table 1).

The decline in suspected cases was further reflected in a persistent decline in the numbers of admitted cases (Fig 1B). The average annual number of cases admitted to the KGH Lassa ward was 148 from 2011 to 2013. During the West African Ebola epidemic an average of 128 cases were admitted. In the three years following the epidemic, an average of 38 cases were admitted (Fig 1B).

## Decline in suspected Lassa fever cases is not due to any major demographic shifts

The decline in suspected Lassa fever cases presenting to KGH did not appear to be due to a major reduction in presentation by any demographic group (Fig 2). However, there were

**Table 1. Characteristics of suspected Lassa fever cases before and after the 2013–2016 West African Ebola outbreak, Kenema, Sierra Leone.** Inclusive of gender, age, survival outcome, and average presentations per year.

| Characteristic | Pre-Ebola[1] n = 2,096 | Post-Ebola[2] n = 918 | P value[3] |
|---|---|---|---|
| Gender, N (%) | | | |
| Male | 855 (41.28) | 429 (46.83) | .005 |
| Female | 1216 (58.72) | 487 (53.17) | |
| Age at presentation, N (%) | | | |
| <5 | 267 (13.22) | 119 (13.05) | < .001 |
| 5–19 | 473 (23.43) | 151 (16.56) | |
| 20–34 | 684 (33.88) | 341 (37.39) | |
| ≥35 | 595 (29.47) | 301 (33.00) | |
| Survival outcome, N (%)[4] | | | |
| Died | 191 (43.31) | 43 (37.39) | .289 |
| Discharged | 250 (56.69) | 72 (62.61) | |
| Average presentations per year, mean (STD) | 698.67 (156.99) | 303.67(42.19) | .049 |

STD = Standard deviation. Differences between overall time period frequencies and characteristic frequencies are due to missing characteristic data.

[1]Pre-Ebola time period defined as 2011–2013.

[2]Post-Ebola time period defined as 20172019.

[3]Fisher's Exact and Kruskal-Wallis tests used to test for general differences between pre-and post-Ebola time periods for categorical and continuous characteristics, respectively,.

[4]Survival outcome at hospital discharge or death observed prior to arrival at KGH. Survival outcomes were unknown for subjects not presenting to KGH unless death was observed prior to presentation.

modest, but perceptible, changes in the demographic make-up of the cases presenting to KGH after the epidemic. From 2011 to 2018, 55–59% of patients evaluated for suspected Lassa fever were women. (Fig 2A). However, in 2019 more males (179) than females (172) presented to KGH. There was a shift in the proportion of female suspected cases from pre-epidemic to post-epidemic of 59% to 55% respectively (p = .005; Table 1).

Age distributions were based on the 2015 Sierra Leone Census and suspected cases were analyzed in four groups: 0–4, 5–19, 20–34, and 35+. There were no major changes in the proportion of all annual suspected cases of Lassa fever associated with each age group, pre- versus post-epidemic (Fig 2B, Table 1). The 20–34 year old age group was the age group with the largest proportion (34%) of suspected Lassa fever cases in the three year period before the West African Ebola epidemic (2011–13) and also during the three year period (2017–19) following the epidemic (37%, Fig 2B, Table 1). However, there was a reduction in the proportion of children and teenagers (5–19 years old) who presented as suspected Lassa fever cases. The decline in the total number of suspected Lassa fever cases pre-epidemic versus post-epidemic among each age group was significant (p < .001, Table 1).

## Fatality rate of suspected cases pre-epidemic compared to post-epidemic

The case fatality rate of suspected cases (CFR; the proportion of deaths to deaths plus discharges) among persons admitted to the KGH Lassa Ward in the three-year period (2017–19) following the West African Ebola epidemic was similar to the CFR in the three-year period prior to the epidemic (2011–2013), 43% and 40% respectively, which was not significantly different (Fig 3, Table 1). There was an increase in CFR among suspected Lassa fever cases during the epidemic (2014–16), which was likely due to patients presenting to the hospital at advanced

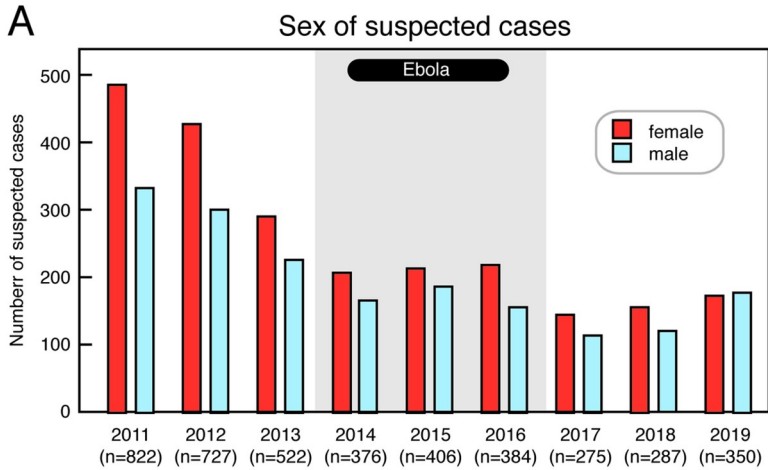

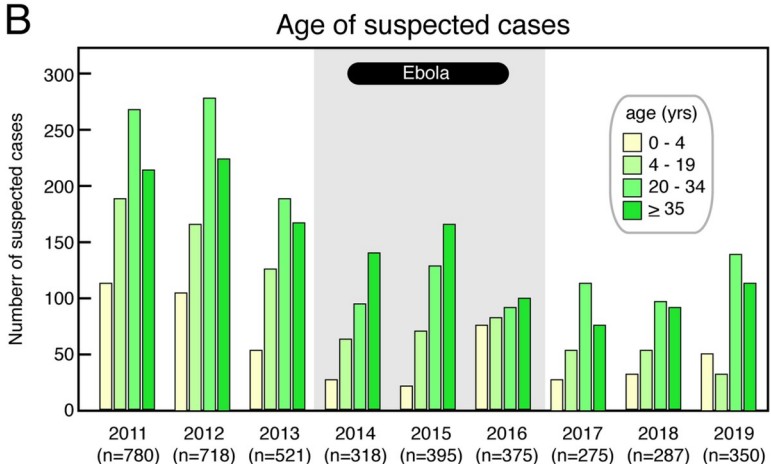

**Fig 2. Sex and age of subjects with suspected Lassa fever presenting to the Kenema Government Hospital.** Panel A: Numbers of female and male subjects presenting with suspected Lassa fever from 2011–19. Panel B: Numbers of subjects presenting with suspected Lassa fever from 2011–19 by age category. Grey shading indicates the years 2014–16 during the West African Ebola epidemic.

stages of illness or with Ebola virus (EBOV). Over the time-period studied, the numbers of deaths and discharges amongst cases admitted to the Lassa Ward declined between the pre-Ebola and post-Ebola periods in parallel with the decline in number of admitted cases.

## Assessment of health seeking behavior questionnaire

The questionnaire was successfully completed for 194 individuals (97%). For six respondents, the full questionnaire was not completed in a timely fashion, and they were excluded from analysis. Of the religions specified by respondents, 89% were Muslim, followed by 11% that were Christian. Of the genders specified by respondents, 66% were female and 34% were male. Of the education levels specified by respondents, 68% received no education, followed by 10% receiving primary education, 20% receiving a secondary education, and 2% receiving a tertiary education (S4 Table). Of female respondents, 16% were currently pregnant, and 87% reported having at least two living children; culturally, given many female respondents had lost children, it was accepted to refer to children that were still alive, whereas medically, any full-term

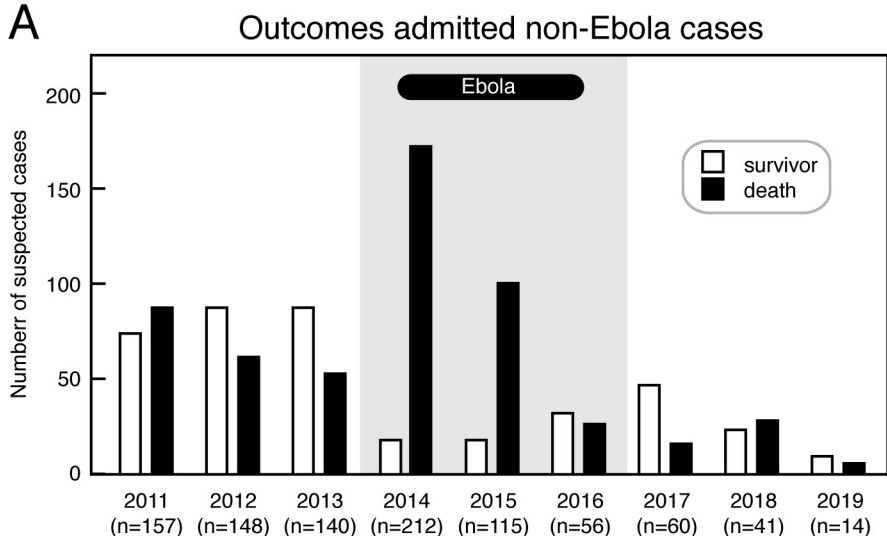

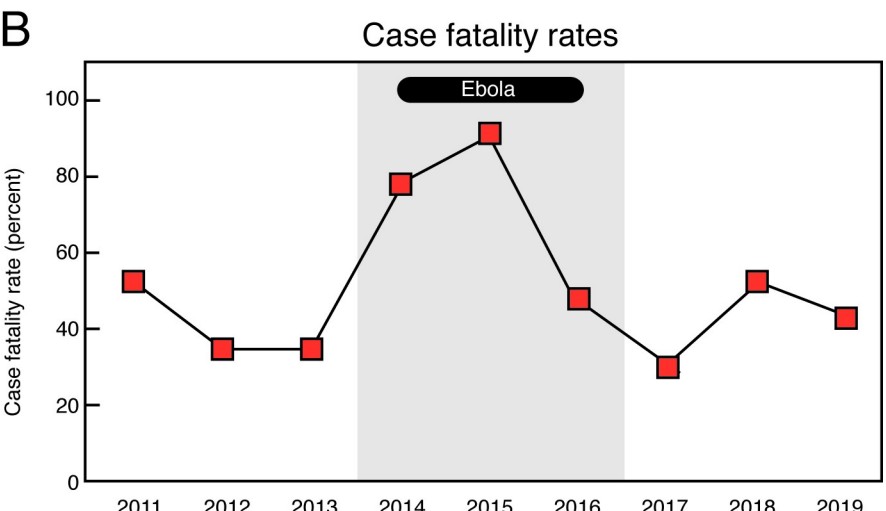

**Fig 3. Discharges, deaths and suspected case fatality rates of subjects with suspected Lassa fever admitted to the Kenema Government Hospital.** Panel A: Discharges and deaths of admitted subjects with suspected Lassa fever from 2011–19. Panel B: Case fatality rates of subjects presenting with suspected Lassa fever from 2011–19. Grey shading indicates the years 2014–16 during the West African Ebola epidemic.

pregnancy may have been relevant Eighty-eight percent of women reported that they go to a government hospital to give birth, and 95% go to a government hospital for postnatal follow up and immunizations (S4 Table).

Analysis of the responses to the questionnaire enabled several features of health seeking behavior (HSB) in Kenema District to be identified (Fig 4, S5 Table). When seeking health care, 91% of respondents reported that they go to a government hospital (Fig 4A). Only 2% reported they used a form of self-treatment or a traditional healer, and 68% of respondents confirmed receiving all their needed services at their health facility (Fig 4B). Moreover, 60% of respondents reported they were likely to seek care within 24 hours of symptoms arising (Fig 4C). According to 87% of respondents, cost was the most common barrier to seeking care at a health facility (Fig 4D). Seventy-two percent of respondents said their attendance at a health-care facility had increased since the epidemic, and 19% said it had decreased (Fig 4E). Seventy-

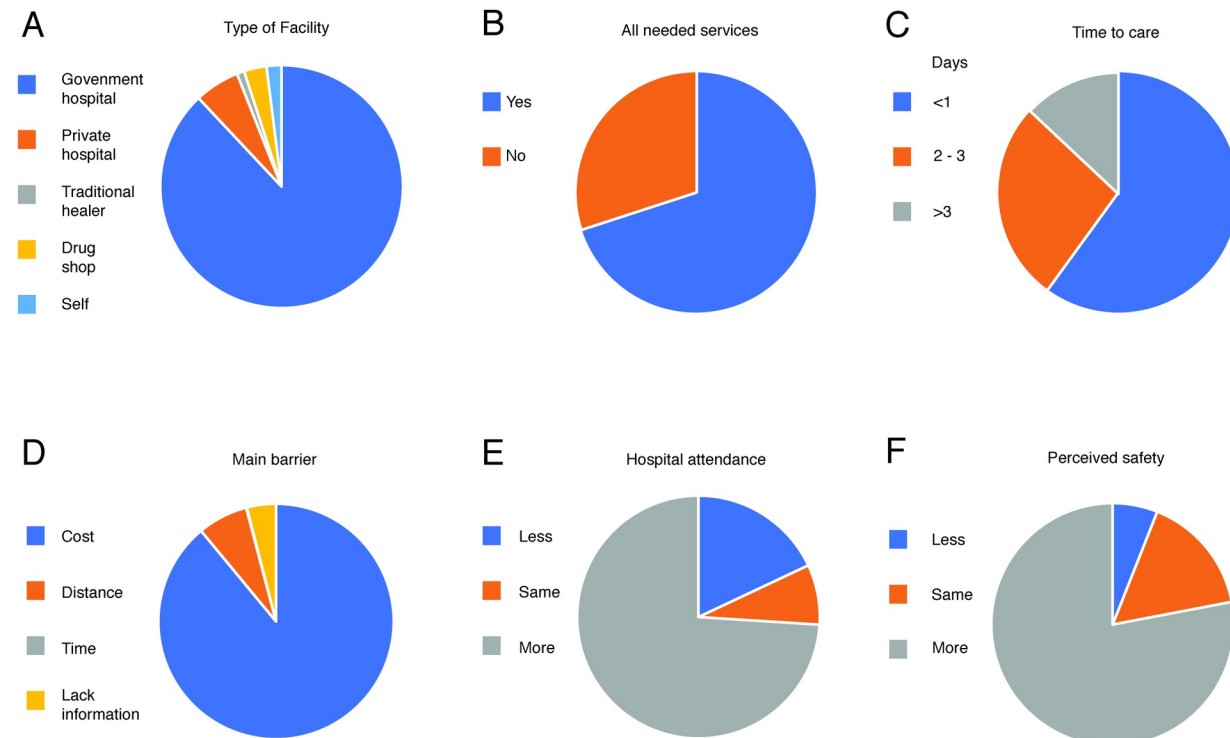

**Fig 4. Features of health seeking behavior by respondents living in Kenema District, Sierra Leone.** Panel A: Type of health facility utilized. Panel B: Did health facility provided all needed services. Panel C: Time lapse until care is sought. Panel D: Main barrier seeking health care. Panel E: Frequency of attendance at a hospital. Panel F: Perceived safety of health facility.

eight percent of respondents stated that they felt that the system was safer post-Ebola epidemic, 5% reported less safe, and 17% said safety of health facilities had remained the same (Fig 4F).

Self-reported satisfaction for health services was higher among males than females ([aOR] = 2.1, [95% CI] = [0.87, 5.10]) (S7 Table). Respondents were likely to say they received all the services they needed at their health facility (S5 Table). There were no other statistically significant differences amongst age groups (with the exception of the 37–45 year age group, but its confidence interval included one); level of education and religious affiliation did not predict answers to either question. There were only three individuals who reported being a survivor of Lassa fever, insufficient for meaningful subgroup analysis.

### Symptoms triggering heath seeking behavior

Responses to the questionnaire were used to identify how likely individuals are to initiate HSB for specific symptoms, both for themselves and their communities (Fig 5). The frequency of symptoms was not determined through the questionnaire and could explain varying levels of HSB by symptom. Fever was the most common symptom prompting individuals to seek care for themselves (Fig 5A). When asked 'which health issues prompt your *community* to seek care for you', fever was also the most common response (Fig 5B). Pain was the next most frequent sign that caused individuals to initiate HSB, followed by headache/nausea, vomiting, and diarrhea. Headache/nausea was the second most frequent health issue noted to prompt community sought care, followed by vomiting, pain, and diarrhea. Bleeding was infrequently reported as a symptom triggering heath seeking behavior in either the individual or community.

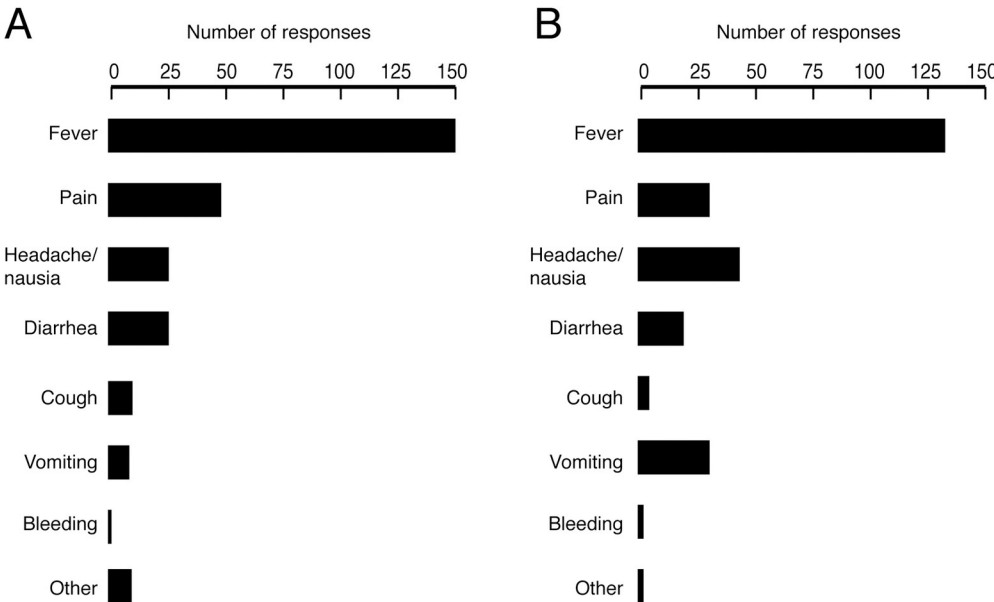

**Fig 5. Reported clinical signs and symptoms initiating health seeking behavior.** Panel A: health issue that would initiate health seeking behavior by the individual. Panel B: health issue that would prompt community to seek care for you.

## Variation in health utilization and satisfaction among villages

The eight villages in which the questionnaire was administered varied in a variety of parameters (S6 Table). The villages examined ranged in size from approximately 600 to 3000 individuals. The villages were 22–51 km away from KGH, with a mean distance from KGH of 38 km. More females answered the questionnaire than males except in villages 1 and 3. The median age of respondents was 35. The level of education varied between villages with a range of 52% to 85% of respondents indicating that they had no formal education. There were statistically significant differences in responses to health utilization and satisfaction among villages (Fig 6). When comparing villages to index village 1, respondents in villages 4 and 6 were less likely to say they got all the services they needed at their health facility (Fig 6A). In response to feelings of safety at hospitals post-epidemic, the majority of residents in all villages said they felt as safe or safer at hospitals (Fig 6B); differences between villages were not statistically significant.

## Discussion

Lassa fever (LF) and Ebola share common characteristics in etiology, clinical presentation, and need for strategic medical intervention such as isolation and intensive hospital care of patients with severe disease. The current study used LF as a metric to assess health seeking behavior (HSB) for febrile illnesses in Kenema District Sierra Leone—a region that was severely impacted by the West African Ebola epidemic. Previously, we reported that the annual numbers of suspected cases seen at Kenema Government Hospital (KGH) from 2008–2012 were similar reflecting the expected "steady-state" incidence for a region with endemic LF [43]. The number of patients evaluated in 2013 for potential LF was consistent with this trend.

Following the 2013–16 West African Ebola epidemic, there has been a notable decline in the number of suspected LF patients presenting to KGH. Potential explanations for the observed decline in Lassa evaluations, include: 1) A decrease in disease incidence, 2) A decrease in population, 3) A change in referral patterns to hospitals and 4) A change in HSB

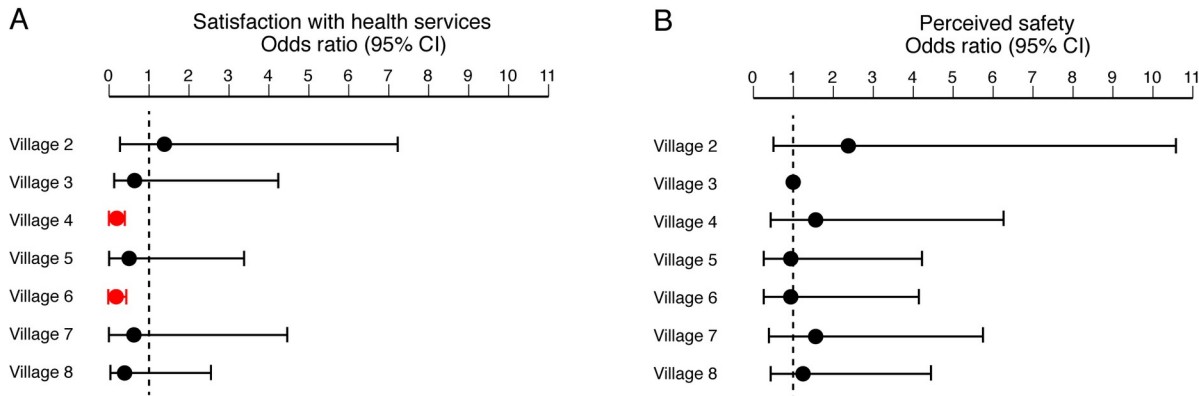

**Fig 6. Features of health seeking behavior by respondents in different villages.** Odds ratios are based off a reference to Village 1. Panel A: Satisfaction with health services. Panel B: Perceived safety of health facility. Red symbols: p < .001.

for febrile illnesses. The evidence indicates that the incidence of LF may actually be increasing in West Africa [47]. The largest outbreaks of LF in Nigeria have occurred in the past three years [48,49]. From 2017–2018, alone, Nigeria experienced their largest ever recorded LF outbreak, which was longer, and included more severe cases than previously seen [50]. Additionally, LF has been detected with increasing frequency in neighboring countries Togo and Benin [51]. While we cannot rule out a localized decrease in Lassa virus (LASV) infection, the incidence of LF is unlikely to have declined by 50% over this brief time frame. Moreover, the population in Sierra Leone has been growing steadily for years. According to census data, since 2012, the average increase in population annually has been 2.3% in Sierra Leone [52]. Outside of the more than 4000 deaths from Ebola, census estimates have indicated ongoing population growth indicating that a population decline substantial enough to explain the dramatic decline in patients is unlikely.

There were changes in the demographics of the suspected LF cases presenting to KGH, including a decrease in the proportion of females and children, including teenagers. However, these demographic changes were modest and do not readily explain the large overall decrease in suspected LF cases. Since the epidemic, considerable resources have helped rebuild the health-care infrastructure at KGH [53]. A new, state of the art Lassa Ward has been built and the research and clinical teams have been fortified. Factors beyond infrastructure constraints are influencing residents seeking care for febrile illnesses. Such changes in HSB for LF are likely multifactorial with almost one in five respondents still reporting not feeling safer in hospital care and nearly 20% of respondents reporting a decreased attendance at hospitals since the epidemic. Given the documented decline in HSB during the epidemic, individuals who reported feeling similar safety levels at hospitals *since* the epidemic are still likely a reflection of a decline in HSB as compared to pre-epidemic levels. As such, for the purposes of this study, they are being considered within the category of individuals who are not reporting increased safety in hospital care.

There have been no programmatic or governmental interventions to drive a change of long existing referral patterns to KGH. A program, started by the government in 2010, to provide free medical care to children under 5 and to pregnant or lactating women has continued [54,55], albeit with major challenges for implementation [56–60]. KGH has served as the regional center of excellence for Lassa care for decades [61]. No other facility has comparable expertise or infrastructure to handle patients with suspected LF. However, other facilities that are capable of testing for LF have recently been established in Sierra Leone including a new

Child/Maternal health hospital operated by Médecins Sans Frontières (MSF)/Doctors Without Borders located outside of Kenema town. The West African Ebola epidemic did impact rural health centers and staff [8,56]. For example, a major health center in Gondama town Bo District, was closed during the epidemic. Prior to the epidemic, Gondama routinely referred suspected Lassa cases to KGH. We did not assess knowledge of LF among rural health care workers (HCW) after the epidemic. Similarly, how the epidemic impacted HCW behavior with regard to LF is unknown. However, a 2017 study assessing community HCW diagnosis in febrile patients in Kenema district found that rapid malaria testing (banned under no touch policies) had been replaced during the epidemic with empiric testing and had not yet returned to pre-epidemic levels [62]. *After the epidemic, HCW's continued to presumptively treat significantly more individuals for malaria than pre-epidemic (p < .001). This change in HCW behavior for febrile illnesses may account for underreporting or decreased recognition of LF patients.* Whether the role and or effectiveness of the rural health centers has changed was beyond the scope of this study.

Given the public health threat Ebola and LF pose, case identification, patient isolation, and vector trapping, or avoidance, are all essential interventions to contain outbreaks. Outreach and educational efforts with local communities and regional health outposts are essential to aid in early case detection to identify at-risk individuals with a febrile illness. Educated providers in health outposts need to identify those with more severe cases to allow transport to better equipped facilities that can provide supportive services and isolation to prevent disease spread. LF poses an ongoing public health threat in West Africa which could be exacerbated if an absence of trust in local or regional medical providers impacts HSB.

Changes in HSB are likely contributing to the notable decline in presentations of suspected LF to KGH. Responses to the HSB questionnaire suggest that confidence in the health infrastructure has not fully rebounded after the epidemic. While the majority of respondents reported hospitals as "safer after the epidemic," and have "increased their attendance at health facilities since the epidemic," around 20% of respondents did not report increased attendance healthcare facilities. This group of respondents, if extrapolated to the regional population, could account for the identified decline in hospital presentation for suspected LF. A persistent wariness, and perhaps distrust, of existing healthcare facilities for care related to possible hemorrhagic fever may remain.

While questionnaire responses varied by village, the sample sizes were too small to identify which specific differences among villages might explain the overall observation. Based on the analysis completed, there were differences among villages in the proportion of respondents that felt they received all the services they needed at their health facility. However, this difference was not explained by gender, religious affiliation, or level of education. Distance from KGH, while considerable, did not explain the different responses. While gender and age have been reported to impact HSB in a variety of studies, we were unable to identify an association given our limited sample size [63–65]. Financial resources of individuals or villages could be a factor, but was not assessed. For each village, neither the incidence of antecedent Lassa illnesses (or death), nor the incidence of Ebola were known. First-hand experience with Ebola or Lassa deaths could influence HSB for febrile illnesses—Ebola survivors are more likely to seek care at a hospital rather than other treatment options [66]. There may be the sentiment that, given the severity of Ebola, LF may be perceived as less severe, associated with a decreased sense of urgency to seek care for symptoms. In conjunction with a potential decrease in concern for the early signs of LF, one can imagine a scenario in which any cost, which is the primary barrier to seeking care reported by 87% of the respondents, would be sufficient to override any inclination to seek medical attention.

We present results from two data sets from two distinct groups of subjects in the Region: one of individuals in the general population and the other of symptomatic patients with suspected LF at KGH. While were unable to draw a direct linkage between the two data sets, we were able to identify some factors that may have facilitated and hindered HSB. Moreover, assessing HSB by an orally administered questionnaire had limitations primarily in sample size and scope. Furthermore, five different individuals delivered the questionnaire and variations in presentation and translation are likely. Bias in self-reporting could further limit the validity of this HSB questionnaire. For example, individuals may report more favorable attitudes towards their local health service for fear that doing otherwise may lead to a decline in service provision. Further understanding the nuances and limitations of the survey can provide additional context; the 78% of respondents reporting 'safer hospitals' and 72% reporting greater 'utilization' could be a reflection of a willingness to seek care for conditions other than suspected hemorrhagic fever. That is, individuals may show increasing HSB for immunizations, maternal/child health care or even HIV or malaria care and remain reticent to seek care for symptoms that may raise concern over a life-threatening illness that mimics Ebola. While it was not within the scope of this project to study, the demand for some health services may rebound more quickly than others. Additionally, the role of local and regional healthcare systems post- the West African Ebola epidemic were not measured in this study. The local and regional healthcare providers serve as an essential bridge to the referral center and may have suffered unrecognized losses from the epidemic that curtail the number of appropriate patients being referred [67]. The need for further research in these areas is essential to understand the factors contributing to HSB in the region and to target interventions to improve the health of these communities long-term.

The goal of this study, to identify and evaluate the persistent impact of the Ebola epidemic on health seeking behavior for LF through hospital level data, and by survey questionnaire of general health seeking behavior, was met given the limitations of the methodology. The study identified a statistically significant decline in the number of individuals presenting to the hospital, or public health unit, with suspected LF. Although limited by sample size, the questionnaire of 194 individuals identified persistent reluctance to seek care for febrile illness since the Ebola epidemic among one fifth of respondents that could account for the decline noted at KGH.

Identifying cause(s) beyond those self-reported for the decline in presentations at the hospital was beyond the scope of our study. What impact changes in community healthcare workers, transportation access, financial constraint or other potential drivers of HSB, had on this decline were not assessed in this study, but likely play an important role. Further research into these, and potentially other, factors influencing underutilization of healthcare for LF is warranted.

To better achieve the goals of this study, a larger sample size, more villages, and more targeted questions regarding the interaction with public health units, and behavior more clearly linked to suspected LF could have strengthened the conclusions. Additionally, targeting multiple aspects of the care pathway would have been valuable. For example, interviewing or analyzing community health workers, observing any changes in cost, distance, or time to care and other barriers identified by respondents in this paper.

After an epidemic, or similar shock to the healthcare system and fabric of society, it is critical to be aware of the impact such an event has or does not have on morbidity, mortality, and healthcare utilization in communities. This research, and its findings, are critical not only for management during an emergency, but in the years following.

This study confirms that health seeking behavior of individuals with suspected LF has not rebounded after the Ebola epidemic despite the maintenance of a state-of-the-art facility.

These findings suggest that to optimize healthcare delivery post epidemic, investments into rebuilding the healthcare system and health of communities need to be multifaceted. Beyond healthcare infrastructure, consideration of other important drivers of health seeking behavior should be considered.

Indirect effects on health and mortality *during* the Ebola epidemic are well known [10–15]. However, factors that may persistently impact outcomes post-epidemic may be more nuanced and less well studied or understood. Monitoring the impact on health seeking behavior (across health indicators) not only during an epidemic but in the years following is important.

Similar to the West African Ebola epidemic, the current coronavirus disease (COVID-19) pandemic is overwhelming healthcare systems, and spreading fear among patients and providers with documentation globally of decreased health seeking behavior for vital health issues such as heart attack, stroke, and deliveries [68]. If, in line with our findings in this paper, HSB decreased during the COVID-19 pandemic and remains below baseline following the pandemic, such information will be vital for appropriate mitigation efforts. Particularly in lower resourced health systems already confronting a sustained impact on health seeking behavior from the West African Ebola epidemic, monitoring how that may be further impacted by COVID-19 will be critical. Optimal health outcomes can only be achieved when individuals seek care appropriately, particularly for infectious diseases that are contagious and often lethal. Improved understanding of the factors that impede HSB is vital to effectively mitigate public health crises as well as help to implement corrective strategies and allow overall health to rebound.

## Supporting information

**S1 Table. Questionnaire in English.** The original, English, version of the questionnaire used in this study to assess reported health seeking behavior in eight villages in Kenema, Sierra Leone.
(DOCX)

**S2 Table. Questionnaire in Krio.** The translated, Krio, version of the questionnaire used in this study to assess reported health seeking behavior in eight villages in Kenema, Sierra Leone.
(DOCX)

**S3 Table. Total suspected cases of Lassa fever, by demographic categories, presenting to Kenema Government Hospital from 2012–2019.** Data collected from the Viral Hemorrhagic Fever Consortium database.
(DOCX)

**S4 Table. Characteristics of Questionnaire Respondents.** Questionnaire respondent characteristics, including gender, religion and educational status.
(DOCX)

**S5 Table. Health Seeking Behavior Questionnaire.** Questionnaire responses for health seeking behavior pertinent questions.
(DOCX)

**S6 Table. Questionnaire demographic responses by village of residence.** Demographic responses to the questionnaire broken down by village of residence. Eight villages in Kenema District are included.
(DOCX)

**S7 Table. Logistic regression assessing satisfaction with health services in 2018 in Kenema district.** Results from a logistic regression analysis assessing satisfaction with health services in Kenema District.
(DOCX)

**S8 Table. Logistic regression results for perceived hospital safety in 2018 in Kenema district.** Results from a logistic regression analysis assessing perceived hospital safety in 2018 in Kenema District.
(DOCX)

## Acknowledgments

We thank the members of the Viral Hemorrhagic Fever Consortium (The Broad Institute, Harvard University, La Jolla Institute of Immunology, the Scripps Research Institute, Tulane University, the University of Texas Medical Branch, Zalgen Labs, the ISTH Lassa Fever Program, and KGH) without which this work could not have been conducted; and the patients with serious febrile illnesses who presented to KGH, as well as their families, without whose cooperation this study would not have been possible. Simbirie C. Jalloh, Allyson M. Haislip, Christopher M. Bishop, Tynette D. Hills, and Douglass Simpson provided program management and logistical support.

## Author Contributions

**Conceptualization:** Mikaela R. Koch, Lansana Kanneh, Paul H. Wise, Lianne M. Kurina, John S. Schieffelin, Jeffrey G. Shaffer, Donald S. Grant.

**Data curation:** Foday Alhasan, Jeffrey G. Shaffer.

**Formal analysis:** Jeffrey G. Shaffer.

**Funding acquisition:** Mikaela R. Koch, Robert F. Garry, John S. Schieffelin, Jeffrey G. Shaffer, Donald S. Grant.

**Investigation:** Mikaela R. Koch, Lansana Kanneh, Foday Alhasan, John S. Schieffelin, Jeffrey G. Shaffer, Donald S. Grant.

**Methodology:** Mikaela R. Koch, Lansana Kanneh, John S. Schieffelin, Jeffrey G. Shaffer, Donald S. Grant.

**Resources:** Robert F. Garry, John S. Schieffelin, Jeffrey G. Shaffer, Donald S. Grant.

**Supervision:** Paul H. Wise, Lianne M. Kurina, Robert F. Garry.

**Visualization:** Robert F. Garry.

**Writing – original draft:** Mikaela R. Koch.

**Writing – review & editing:** Mikaela R. Koch, Robert F. Garry, Jeffrey G. Shaffer, Donald S. Grant.

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
