## [Decision Letter · Decision Letter 0]

24 Dec 2020

Dear Dr. Shaffer,

Thank you very much for submitting your manuscript "Health seeking behavior after the 2013-16 Ebola epidemic: Lassa fever as a metric of persistent changes in Kenema District, Sierra Leone" for consideration at PLOS Neglected Tropical Diseases. As with all papers reviewed by the journal, your manuscript was reviewed by members of the editorial board and by several independent reviewers. In light of the reviews (below this email), we would like to invite the resubmission of a significantly-revised version that takes into account the reviewers' comments. 

We cannot make any decision about publication until we have seen the revised manuscript and your response to the reviewers' comments. Your revised manuscript is also likely to be sent to reviewers for further evaluation.

Sincerely,

Ayato Takada, Ph.D.

Deputy Editor

Ayato Takada

Deputy Editor

Reviewer's Responses to Questions

**Key Review Criteria Required for Acceptance?**

**Methods**

-Are the objectives of the study clearly articulated with a clear testable hypothesis stated?

-Is the study design appropriate to address the stated objectives?

-Is the population clearly described and appropriate for the hypothesis being tested?

-Is the sample size sufficient to ensure adequate power to address the hypothesis being tested?

-Were correct statistical analysis used to support conclusions?

-Are there concerns about ethical or regulatory requirements being met?

Reviewer #1: The methodology is well written and explains well the details of data collection. While the total sample population size is not huge for a study such as this, the inclusion multiple villages populations increases the strength of this paper. I have no concerns regarding the ethical or regulatory requirements being met.

Reviewer #2: Yes, study objectives were clearly stated, with appropriate design and analysis to support conclusion. Sample size was low but limitation clearly acknowledged. Ethical requirement needed for the study was adequate.

Reviewer #3: • Line 188 – “rates was,” – I believe the comma should be placed after “rates” instead of “was”.

• Line 194 – “VHFC” – This abbreviation is used here without the full term having appeared in the text. Please include this prior to the abbreviation.

• Lines 202-204 – “Fisher’s Exact and Kruskal-Wallis tests were used to compare categorical and continuous characteristics, respectively, between pre-and post-Ebola time periods. Significance tests were performed using the SAS statistical application.” – It would aid the reader in interpreting the results if the authors could provide more details of the significance tests performed. For example, the meaning of a statistically significant p-value in the specific context of the two tests used.

• Lines 206-207 – “HSB responses were analyzed to identify potential predictors of responses to the Lassa pertinent questions” – Does this statement mean HSB responses were analysed to determine predictors for the specific responses to the Lassa fever related questions? Please reword to improve clarity of meaning.

• Line 207 – “The analysis was performed by running logistic regression models in STATA” – Could you please provide more specific details of your logistic regression models, including the selection process of variables used in the final model?

**Results**

-Does the analysis presented match the analysis plan?

-Are the results clearly and completely presented?

-Are the figures (Tables, Images) of sufficient quality for clarity?

Reviewer #1: In general, the results are clearly written and easy to follow, although I do feel as if several of the figures could be displayed in different ways. For example, given that all the panels of Figure 4 could be represented as pie charts to more clearly represented the proportions (although this is just the reviewer’s preference, they are fine as they are!).

Reviewer #2: Yes, analysis presented correspond with plan and results and figures were clearly presented.

Reviewer #3: • Line 214 – “In the three-year period (2017-19) following the West African Ebola outbreak, there was a decline…” – Please revise this wording because it currently implies a decline over the period of 2017-2019, rather than compared to the previous 6 years. Fig1A shows a slight increase from 2017 to 2019.

• Line 222 – “(p = .034, Table 1)” – The p-value here is inconsistent with the value in Table 1 (0.049). Please review and remove any inconsistencies.

• Line 259 – “However, in 2019 more males than females presented to KGH.” – It would be more informative to the reader if the value difference between female and male presentations was provided here.

• Line 261 – “(p = .029; Table 1)” – The p-value here is inconsistent with the value in Table 1 (0.005). Please review and remove any inconsistencies.

• Lines 275-276 – “The decline in suspected cases remained statistically significant across all age groups (Fig 2).” – Please consider revising this sentence to improve clarity of meaning. For example, it should be made explicit if this is comparing the three years pre-Ebola to the three years post-Ebola. Furthermore, Figure 2 does not present any data regarding the statistical significance of the change in number of suspected cases per age group, therefore I suggest the authors either remove this claim of statistical significance or substantiate it via other evidence presented in the paper.

• Lines 276-277 and lines 282-283 – “There were no major changes in the proportion of suspected Lassa fever cases in most age groups pre- vs post- epidemic” and “The differences pre-epidemic and post-epidemic in the age group proportions among suspected Lassa fever cases was significant” – These two claims are contradictory to each other. Please revise the text to remove this inconsistency and more clearly state the results in the table.

• Line 273 – Figure 2B – The legend for age groups shows the category of 4-19 years, but the text specifies the age range of 5-19 years. Please revise this as appropriate.

• Lines 277-278 – “The 20-24 year old age group was the largest proportion…” – I believe this should be the “20-34” year old age group. Furthermore, I would recommend rewording slightly to say that “The 20-34-year-old age group was the age group with the largest proportion…”

• Line 280 – “(35%, Fig 2B, Table 1)” – The proportion for 20-34 in Table 1 is recorded as 37%. Please review and correct as appropriate to remove any inconsistency.

• Lines 280-281 – “…there was a reduction in the number of children and teenagers…” – Please consider rewording this sentence to make it explicit that it is the reduction in the proportion of suspected cases being referred to and not simply the raw number.

• Line 286 – “The suspected case fatality rate…” – It may improve the readability of this sentence to rephrase this as “The case fatality rate of suspected cases…”. Additionally, subsequent uses of “suspected CFR” should be similarly reworded or simply be replaced with “CFR”.

• Lines 287-289 – “the West African Ebola epidemic” and “the Ebola outbreak” – Please consider rewording to be consistent in term use. Throughout the text, the latter of these appears to be more frequently used.

• Lines 292-294 – “Over the time-period studied, the numbers of deaths and discharges amongst cases admitted to the Lassa Ward consistently declined in parallel to the numbers of overall suspected cases.” – It is unclear what is meant by “consistently declined in parallel”, when the numbers presented in Figure 3A are based on the same numbers as Figure 1B. Is this stating that the three-year-average of both discharges and deaths has fallen between the pre-Ebola period and the post-Ebola period in the same way that the three-year average fell between these two periods for number of admitted cases? Please reword this sentence to improve clarity of meaning.

• Line 307 – Figure 3B – This figure appears to be incomplete. It only shows a single line with red markers for values, but it is described as the fatality rate “by age category”. Please revise such that this figure shows the data for all age categories.

• Lines 311-312 – “Of those completing the questionnaire, 89% were Muslim, 66% were female, and 69% had no education.” – Please consider revising this summary of results such that each category of results is provided in its own context, e.g. Of the religions specified by the respondents, 89% were Muslim, followed by…

• Line 313 – “The majority of women reported having two living children.” – Please include the percentage value for this majority. Furthermore, the term “living children” is ambiguous. Please consider rewording to be more precise.

• Lines 313-315 – “88% of women…” – Please spell out the percent value here as sentences should not start with a number value.

• Lines 317-318 – “When seeking health care, 89% of respondents reported that they go to a government hospital” – This value is reported as 90% in S5 Table. Please review and correct the inconsistency as appropriate.

• Lines 318-319 – “Only 4% reported they used a form of self-treatment or a traditional healer” – The values in S5 Table for self-treatment (0%) and traditional healer (1.0%) do not add up to 4%. Please review and correct the inconsistency as appropriate.

• Lines 319-320 – “68% of respondents confirmed receiving all their needed services at their health facility” – While shown in Figure 4B, this information is not presented in S5 Table. Please ensure that all survey answers are presented in the corresponding tables in the supplementary text.

• Lines 320-321 – “individuals reported they were likely to seek care 24 hours after symptoms arose…” – According to the survey questions listed in the supplementary text, this should be “within 24 hours”.

• Lines 321-322 – “As noted by 87% of respondents, cost was the most common barrier preventing subjects from seeking care at a health facility” – Please consider an alternative expression to “As noted by”, such as “According to”. Furthermore, please consider replacing “subjects” with “respondents” to increase consistency.

• Lines 322-324 – “72% of respondents said their attendance at a healthcare facility had increased since the epidemic, and 19% said it had decreased” - While shown in Figure 4E, this information is not presented in S5 Table. Please ensure that all survey answers are presented in the corresponding tables in the supplementary text.

• Lines 324-325 – “78% of respondents stated… …remained the same” - While shown in Figure 4F, this information is not presented in S5 Table. Please ensure that all survey answers are presented in the corresponding tables in the supplementary text.

• Lines 339-340 – “There were no significant differences in responses to health utilization or perceived safety of hospitals between men and women, or distance from KGH (S7 Table, S8 Table).” – S7 Table’s label suggests it describes satisfaction with health services rather than utilization. Please revise to correct this inconsistency. Furthermore, these two tables only present OR and 95% CI. Please consider including the p-value to these tables to better substantiate claims of statistical significance. 

• Lines 340-341 – “With the exception of a group without a recorded age (group 5) respondents were likely to say they received all the services they needed at their health facility (S8 Table).” – This is the first use of a “group” in the text and serves no meaningful purpose. If this term is deemed necessary, please define it in the Methods section of the manuscript. Furthermore, S8 Table only lists four categories for age. Please review and revise as appropriate.

• Providing the p-values specific to these statements in the main text will also add to the reader’s understanding of the results.

• Lines 348-355 – “Symptoms triggering heath seeking behavior” – The interpretation of these results is over-simplifying the frequency of symptoms because it only demonstrates whether the occurrence of symptoms leads to HSB, but does not quantify the frequency of each symptom. For example, the low frequency recorded for bleeding may be more attributable to the survey participants simply not having experienced those symptoms rather than them not seeking health care for that specific symptom. Without quantitative data on frequency of symptoms, it is not possible to fully describe which symptoms are most and least likely to lead to HSB. Therefore, care should be taken to not overstate the meaning of the recorded responses. This lack of quantitative data should be discussed as one of the limitations of the study.

• Lines 365-377 – “Variation in health utilization and satisfaction among villages” – It is not clear why Village 1 was chosen as a reference village for this village level analysis. This should be clearly described in the Methods section. 

• Lines 374 – “…respondents in villages 3 and 6…” – based on Figure 6A I believe this should be villages 4 and 6. Please review and revise as appropriate.

**Conclusions**

-Are the conclusions supported by the data presented?

-Are the limitations of analysis clearly described?

-Do the authors discuss how these data can be helpful to advance our understanding of the topic under study?

-Is public health relevance addressed?

Reviewer #1: The discussion is well written, discusses the wider relevance of the data presented in the results, and brings in valid discussion points as to the relevance of the work carried out.

Reviewer #2: The conclusion of the study was supported by the data presented and analysis made.

Reviewer #3: • Line 397 – “There is no evidence that the incidence of Lassa fever is decreasing in West Africa” – Please provide a reference immediately following this statement.

• Line 399 – “…in Togo and Benin [12]” – Please check that this is the correct reference and replace if necessary.

• Line 413 – “one in five respondents still reporting not feeling safer in hospital care” – The group of respondents who specified “same” for their feeling of safety should not be included as possible cause for decline in HSB for Lassa fever. Please revise this statement to only include those who answered “less”. Please also considering revising the statement on lines 450-451 similarly.

• Lines 450-453 – “…around 20% of respondents… …non_lassa visits.” – The connection being drawn between a lack of increase in trust in healthcare facilities and a 20% decline in presentation for suspected Lassa fever at KGH is unclear. Please revise and elaborate on this discussion and further describe how the authors drew this connection from the analysis.

• Line 451 – “Given that individuals do not present more than once with Lassa fever” – Please provide a reference for this claim.

• Lines 465 – “One can imagine… …of Lassa fever.” – Please revise this statement to increase clarity of meaning, in particular the meaning of “override a decreased concern”.

**Editorial and Data Presentation Modifications?**

Reviewer #1: None

Reviewer #2: (No Response)

Reviewer #3: General

• Throughout the paper, where percentage values are presented, I would recommend providing at least 2 decimal places of precision. Please ensure this is consistently applied to all data in both the main manuscript and the supplementary text.

• Abbreviated terms should be explicitly stated in full for their first use in each major section, i.e. Introduction, Methods, Results, Discussion. For example, HCW is defined as health care workers in the Introduction, but not in the Discussion section.

• A number of minor spelling and grammar mistakes exist through the paper, please proofread the entire manuscript to correct these.

• The Methods section does not provide sufficient details describing why the authors adopted the statistical approaches used in this study. Furthermore, the hypotheses being tested in this study are not made clear, which makes it difficult to accurately interpret the results.

• The Results section contains incomplete figures, incorrect references to values in tables, and unclear descriptions of some of the results recorded.

• The Discussion section presents a wide variety of topics related to HSB, however, there is very limited discussion specific to the findings of this study. Furthermore, there is no discussion of the results of the logistic regression analyses performed in the study. The discussion also comments that the individual village sample sizes were too small for inter-village comparison, however these comparisons are still presented in the results with no further elaboration in the Discussion section.

• S1 and S2 Tables – In both languages for the questionnaire, there is a note next to question 2 to skip to question 8 if the interviewee is male, which would skip over questions relating to Lassa fever history, education, and religion, but the numbers presented in S3 Table suggest that these questions were answered by both male and female participants. Could you please review the questionnaires included in the supplementary text to ensure they accurately represent the actual questionnaire performed based on the results presented.

• S4 Table lists the numbers of respondents per question, but some of these are inconsistent. For example, level of education is listed as being answered by 160 participants in the note at the bottom of the table, but the numbers within the table sum to 191. Furthermore, the main text on lines 311-312 references the percentage value for no education of all who completed the questionnaire rather than of the 191 who answered this question, without any mention of the different questions of the survey being answered by varying numbers of participants.

• S7 and S8 tables list the age categories 28-36, 37-45, 45+, and missing, whereas the main text uses the categories <5, 5-19, 20-34, and 35+, but there is no explanation for this difference in categorisation of ages. Please review this different categorization and provide an explanation in the main text when the age categories from these tables are first referenced.

Abstract

• Line 39 – It is stated that “dramatic underreporting and substantial declines in hospital presentations”, but the scale of this change is not clear. It would assist in understanding the magnitude of the change in hospital presentations if some metric were used to compare pre- and post-Ebola outbreak.

Author Summary

• Line 76 – “a questionnaire given to nearly 200 area residents” – The term “area residents” is ambiguous. Please consider alternate wording. Furthermore, the abstract states that 200 surveys were given but 194 (nearly 200) were completed. Please consider revising wording to more accurately describe the numbers presented.

Introduction

• Line 96 – “While Ebola was directly killing thousands” – The expression “directly killing thousands” could be better reworded as “directly responsible for the deaths of thousands”. Additionally, it may be helpful to provide more specific figures in this sentence (perhaps contained in the references), as the term “thousands” is used twice in the sentence, which does not provide a clear sense of relative scale.

• Line 96 – “…changes in health seeking behavior (HSB) due to…” – Given that HSB is a key topic of this paper, it would be beneficial to the reader for the authors to provide their own definition of HSB. Furthermore, the Introduction later states that a goal of the study was to quantify HSB, therefore it would be beneficial to provide examples of how HSB can be quantified along with this definition of HSB.

• Line 98 – “At the height of the epidemic, there was a 31- 37% drop in facility-based deliveries” – The references show that this sentence is referring to child delivery but manuscript text should be more explicit. Furthermore, it is not made clear how this directly relates to the previous statement regarding additional deaths due to changes in HSB. It may be beneficial to the reader to add some additional context such as what impact the changes in HSB have had on maternal health. Additionally, the geographical scope of this figure is unclear. Is the drop in deliveries that of Sierra Leone or a region therewithin?

• Lines 103-104 – “the attitudes, beliefs and perceptions of the healthcare system” – Please consider revising the word ordering of this text to improve readability. 

• Lines 104-105 – “In the wake of the Ebola epidemic, there has been significant investment and attention in health systems strengthening for the affected countries of West Africa” – It would be beneficial to the reader if some specific examples of how the health system has been strengthened were provided, particularly relating to Sierra Leone.

• Lines 107-108 – “HSB of patients with suspected Lassa fever” – The abstract did not specify that the recipients of the survey were patients with suspected Lassa fever; it only states that residents from 8 villages were surveyed. Please consider rewording to increase consistency throughout the manuscript.

• Lines 113-114 – “While many individuals infected with LASV do not present to a health care provider or facility” – It is not clear if this is referring to the situation prior to or during/after the Ebola outbreak. The geographical scope of the statement is also unclear (i.e. Sierra Leone or Western Africa). Please reword to make this more explicit.

• Line 117 – “the Mastomys natalensis” – Please consider removing the “the” from this species reference.

• Line 121 – “hemorrhage” – I think this should be “hemorrhaging”.

• Line 124 – Please confirm that reference #9 is the correct reference for this statement. Furthermore, the style of reference numbering has changed from parentheses to square brackets here and in the following line. Please revise for consistency.

• Line 124 – The abbreviation “LF” is used here for the first time. It should be first used immediately following the first use of the full expression “Lassa fever”.

• Lines 127 – “quantify HSB for suspected Lassa fever” – Please revise this text to clarify the intended meaning. Furthermore, the abstract suggests that study’s aim was to assess the impact of the Ebola outbreak on HSB, suggesting that the goal here should in fact be to quantify the change in HSB due to the Ebola outbreak. Please review and revise as necessary.

• Lines 128-129 – “an impact on HSB due to the Ebola epidemic” – This may be easier to read when worded as “the impact of the Ebola epidemic on HSB”.

• Lines 129-130 – “Lassa endemic” – Please consider rewording to “Lassa fever endemic”.

• Line 130 – “identify barriers” – Please consider rewording and/or providing examples to make meaning of “barriers” clearer.

**Summary and General Comments**

Reviewer #1: There is a lot of useful information in this manuscript, and the data will be useful in planning for both standard year-to-year LASV epidemiology and diagnostics studies, as well as help shape future public health strategies in the region. The report is well written and clearly addresses a gap in the data available from this region, especially regarding the trust in HCW and the health care infrastructure following the 2013-2016 EBOV epidemic. The main recommendations are edits to the tables and figures. Please see specific comments below. 

General Comments:

Statistical strength of the data is mentioned numerous times in the manuscript, with references to the relevant figures. However, there is no visual representation of statistics on the figures which would help the reader quickly understand the differences between data groups. If the authors could find a way to add representations of significance to the figures I think this would improve the manuscripts readability.

Specific Comments:

I think it would be useful for the reader to have a map of the area sampled, so the reader could have an understanding of where villages 1-7 are located both in relation to Kenema and in relation to Sierra Leone as a whole.

Table 1: I’m unclear as to what line the p values are referring to? There seems to be one value per characteristic ,although each characteristic has multiple criteria? 

Lime 374: Should be villages 4 and 6?

Reviewer #2: (No Response)

Reviewer #3: • The concept of this paper is novel and presents a subject of high importance to Ebola affected countries such as Sierra Leone. 

• There is only limited literature currently available on HSB in West Africa, and especially Kenema, Sierra Leone, therefore the concept of this paper is quite novel, which will hopefully assist in the development of targeted interventions that lead to improved HSB amongst residents, not only in Kenema, but across Sierra Leone.

• The paper makes it clear that presentations for Lassa Fever at KGH have declined post-Ebola outbreak compared to pre-Ebola outbreak, however, the connection between change in HSB and this decline in presentations is not made clear by the analysis presented, therefore the evidence and analysis provided do not sufficiently assess the impact of the Ebola outbreak on HSB with respect to presentations to KGH.

• The discussion does not fully evaluate how well the goals of the study have been achieved and provide detailed suggestions of how to better achieve the goals which were not fully addressed by the study.

• The paper does not provide sufficient discussion of how this study’s findings can be used to develop targeted public health strategies, particularly in Kenema.

Other comments from Reviewer #2.

Line 390: “series” is not clear and should be reworded.

Lines 490-494: These sentences should be revised for more clarity.

PLOS authors have the option to publish the peer review history of their article (what does this mean?). If published, this will include your full peer review and any attached files.

Reviewer #1: No

Reviewer #2: No

Reviewer #3: No
---

## [Decision Letter · Decision Letter 1]

10 May 2021

Dear Dr. Shaffer,

Thank you very much for submitting your manuscript "Health seeking behavior after the 2013-16 Ebola epidemic: Lassa fever as a metric of persistent changes in Kenema District, Sierra Leone" for consideration at PLOS Neglected Tropical Diseases. Your manuscript was re-reviewed by members of the editorial board and by the same reviewers. The reviewers appreciated the attention to an important topic. Based on the reviews, we are likely to accept this manuscript for publication, providing that you modify the manuscript according to the review recommendations. 

Sincerely,

Christopher M. Barker

Associate Editor

Ayato Takada

Deputy Editor

Reviewer's Responses to Questions

**Key Review Criteria Required for Acceptance?**

**Methods**

-Are the objectives of the study clearly articulated with a clear testable hypothesis stated?

-Is the study design appropriate to address the stated objectives?

-Is the population clearly described and appropriate for the hypothesis being tested?

-Is the sample size sufficient to ensure adequate power to address the hypothesis being tested?

-Were correct statistical analysis used to support conclusions?

-Are there concerns about ethical or regulatory requirements being met?

Reviewer #1: (No Response)

Reviewer #3: No comments for this section.

**Results**

-Does the analysis presented match the analysis plan?

-Are the results clearly and completely presented?

-Are the figures (Tables, Images) of sufficient quality for clarity?

Reviewer #1: (No Response)

Reviewer #3: • Lines 337-338 – “Moreover, individuals reported they were likely to seek care within 24 hours of symptoms arising (60%; Fig 4C).” – I recommend rewording such that the percentage is stated within the sentence to be consistent with the surrounding text. Furthermore, I recommend replacing “individuals” with “respondents”, i.e 60% of respondents.

• Lines 339-341 – “72% of respondents…” and “78% of respondents…” – Please replace numerical figures with spelled out numbers when at the start of a sentence.

• Lines 350-351 – “Self-reported satisfaction for health services was significantly higher among males than females ( [aOR] =2.1, [95% CI] = [0.87, 5.10]) (S7 Table, S8 Table).” – I recommend that the authors remove the expression “significantly” because the p-value for this result in S7 table is not statistically significant. Furthermore, this statement does not discuss perceived safety and therefore no reference to S8 Table is required.

• Lines 351-352 – “Respondents were likely to say they received all the services they needed at their health facility (S8 Table).” – The results described by this sentence are not contained in S8 Table. Please revise the text and/or reference as necessary.

• Lines 352-354 – “There were no other statistically significant differences amongst age groups; level of education and religious affiliation did not predict answers to either question.” – S8 Table shows a p-value of less than 0.05 for the age group 37-45. Please revise the text accordingly to reflect the presence of this statistically significant variable.

• Lines 355-357 – “For this analysis, age categories, 18-27, 28-36, 37-45, and 45+, were selected to accommodate the age inclusion criteria for the HSB questionnaire.” – This information would be more appropriately included in the Methods section.

**Conclusions**

-Are the conclusions supported by the data presented?

-Are the limitations of analysis clearly described?

-Do the authors discuss how these data can be helpful to advance our understanding of the topic under study?

-Is public health relevance addressed?

Reviewer #1: (No Response)

Reviewer #3: • Lines 409-410 – “Nigeria experienced their largest ever recorded LF outbreak” – Lassa fever is used in its abbreviated form here for the first time in the Discussion section with no prior definition. Furthermore, it is used in its full form several times prior to this sentence within the Discussion section. Please define after the first use and use the abbreviated version for subsequent uses.

• Line 426 – “one in five respondents” – I suggest adding “almost” to this statement because the actual figure is 22% according to S5 Table.

• Lines 449-450 – “Instead, HCW’s empirically treated statistically significantly more individuals for malaria post- vs. pre- epidemic.” – The meaning of this is not clear. If this is presenting a comparison to the previous sentence’s statement then please more explicitly describe this connection. Furthermore, it would be more meaningful to provide the numeric value of this statistical significance.

• Lines 468-469 – “…around 20% of respondents did not report increased trust in healthcare facilities. This group of 20%...” – I think it would be more accurate to directly refer to the survey answer of decreased attendance at Kenema hospital after the 2014 Ebola epidemic rather than the interpreted meaning of “increased trust in healthcare facilities”. While this connection may be possible, the survey did not directly ask respondents about their trust in healthcare facilities. Furthermore, I would recommend replacing “This group of 20%” with “This group of respondents” because the group is not exactly 20%.

• Lines 479-481 – “While others have reported differences in health seeking behavior among genders and age groups, our limited sample size was unable to identify an association.” – Please be more explicit as to what is meant by “others”. If this is referring to other studies then please provide supporting references.

• Line 481 – “Economic strength could be a factor, but was not assessed.” – It is unclear what is meant by “economic strength”. Please revise this sentence to better explain this factor.

• Lines 492-496 – “A limitation of these results is that two data sets for different groups of subjects were used: one for asymptomatic individuals, and the other for symptomatic suspected Lassa fever cases. HSB cannot truly be measured until an individual experience’s symptoms. As such, we were unable to draw a direct linkage between the two datasets and the authors relied primarily on the surveillance data to draw conclusions.” – The intended meaning of this paragraph is difficult to understand. I think that the fact that the study used data from two different sources is not a limitation in itself. While a direct linkage between the decline in LF presentations at KGH and changes in HSB could not be sufficiently assessed, this study was able to explore and identify some factors that led to initiation of HSB and factors that acted as barriers to HSB. Please revise this paragraph.

• Line 550 – “Indirect effects on health and mortality during an epidemic are well known.” – Please add supporting references for these well known effects.

**Editorial and Data Presentation Modifications?**

Reviewer #1: (No Response)

Reviewer #3: Introduction:

• Lines 112-114 – “Commonly used proxies for health seeking behavior have been healthcare consumption patterns, provider visits, and self-reported care seeking.” – Please provide a reference to support this statement.

• Lines 123-125 – “This study sought to comprehend that knowledge gap by 1) assessing HSB of patients with suspected Lassa fever and 2) assessing general HSB of asymptomatic individuals through use of a questionnaire, in the Kenema district of Sierra Leone, a region that was severely impacted by Ebola.” – From my understanding after reading the entire manuscript, the HSB questionnaire was administered to 200 Sierra Leoneans across 8 villages in the Kenema district; however, the manuscript does not describe any specific selection of asymptomatic individuals for the survey. There may in fact have been asymptomatic individuals amongst those who participated the HSB survey; however, I would suggest removing the term “asymptomatic” from this sentence to avoid confusion.

**Summary and General Comments**

Reviewer #1: All questions and comments addressed. Manuscript is now much stronger.

Reviewer #3: Thank you for responding to all of the comments raised in my initial review.

Compared to the original draft, the authors have made significant improvements to the quality and presentation of their manuscript. The concept of this paper is novel and interesting. However, there are still a number of finer points requiring further elaboration and clarification.

PLOS authors have the option to publish the peer review history of their article (what does this mean?). If published, this will include your full peer review and any attached files.

Reviewer #1: No

Reviewer #3: No

Figure Files:

Data Requirements:

Reproducibility:

References

---

## [Editor Report · Decision Letter 2]

19 Jun 2021

Dear Dr. Shaffer,

We are pleased to inform you that your manuscript 'Health seeking behavior after the 2013-16 Ebola epidemic: Lassa fever as a metric of persistent changes in Kenema District, Sierra Leone' has been provisionally accepted for publication in PLOS Neglected Tropical Diseases.

Best regards,

Ayato Takada, Ph.D.

Deputy Editor

Ayato Takada

Deputy Editor

---

## [Editor Report · Acceptance letter]

7 Jul 2021

Dear Dr. Shaffer,

We are delighted to inform you that your manuscript, "Health seeking behavior after the 2013-16 Ebola epidemic: Lassa fever as a metric of persistent changes in Kenema District, Sierra Leone," has been formally accepted for publication in PLOS Neglected Tropical Diseases.

Best regards,

Shaden Kamhawi

co-Editor-in-Chief

Paul Brindley

co-Editor-in-Chief
